# Necessity of Pelvic Lymph Node Irradiation in Patients with Recurrent Prostate Cancer after Radical Prostatectomy in the PSMA PET/CT Era: A Narrative Review

**DOI:** 10.3390/biomedicines11010038

**Published:** 2022-12-24

**Authors:** Naeim Nabian, Reza Ghalehtaki, Felipe Couñago

**Affiliations:** 1Radiation Oncology Research Center, Cancer Research Institute, Tehran University of Medical Sciences, Tehran P.O. Box 1419733141, Iran; 2Department of Radiation Oncology, Cancer Institute, Tehran University of Medical Sciences, Tehran P.O. Box 1419733141, Iran; 3Department of Radiation Oncology, San Francisco de Asís and La Milagrosa Hospitals, GenesisCare, 28010 Madrid, Spain

**Keywords:** prostatic neoplasms, recurrence, lymph nodes, PSMA PET/CT, gallium 68 PSMA-11, prostate cancer, radiation therapy/radiotherapy, pelvic LN

## Abstract

The main prostate cancer (PCa) treatments include surgery or radiotherapy (with or without ADT). However, none of the suggested treatments eliminates the risk of lymph node metastases. Conventional imaging methods, including MRI and CT scanning, are not sensitive enough for the diagnosis of lymph node metastases; however, the novel imaging method, PSMA PET/CT scanning, has provided valuable information about the pelvic LN involvement in patients with recurrent PCa (RPCa) after radical prostatectomy. The high sensitivity and negative predictive value enable accurate N staging in PCa patients. In this narrative review, we summarize the evidence on the treatment and extent of radiation in prostate-only or whole-pelvis radiation in patients with positive and negative LN involvement on PSMA PET/CT scans.

## 1. Development of PSMA PET/CT Scan

The conventional imaging methods, such as magnetic resonance imaging (MRI) and computed tomography (CT) scanning, have limited diagnostic accuracy for lymph node (LN) involvement in patients with prostate cancer (PCa) since such methods are dependent on size and basic morphological criteria for the diagnosis of LN involvement [1]. Accordingly, pelvic lymph node dissection (LND) is considered the gold standard in LN staging. However, not all patients are candidates for or elect to undergo operative management. Moreover, surgery is also associated with a dramatic risk of complications, considering the relatively old age of the affected patients [2,3]; hence, an accurate noninvasive imaging technique is required to overcome these limitations. 

Over the past few years, the development of prostate-specific membrane antigen (PSMA) positron emission tomography (PET) has transformed the diagnosis and management of PCa. This revolution goes back to 1996, when N-Acetylated alpha-linked acidic dipeptidase was introduced as a high-affinity agent that was primarily used for the treatment of neurologic disorders [4] and was further used for PET imaging of the brain to study glutamatergic transmission [5]. In the following years, carbon-11 (^11^C)-choline, fluorine-18 (^18^F)-fluoricholine, C-acetate, fluciclovine, gallium-68 (^68^Ga), and F-radiolabeled chemicals were introduced as PET scan agents and gained popularity for a short period worldwide. This was because of their advantage in directly imaging cancer, rather than the surrounding bone, compared to bone-directed PET agents [6,7,8]. Although the diagnostic accuracy of these PET agents in the diagnosis of prostate cancer has been confirmed, variable sensitivity and specificity values have been reported regarding patient-related, cancer-related, and treatment-related factors [8]. In a comparison of the diagnostic accuracy of the two agents approved in the United States, C-choline and fluciclovine, the latter appeared to detect more local, nodal, and bony diseases with a higher sensitivity (37% vs. 32%) and specificity (67% vs. 40%) [9].

The first practical human PSMA PET agent, ^18^F-DCFBC, was developed in 2008. This type II transmembrane protein, in human zinc-containing metalloenzyme with glutamate carboxypeptidase/folate hydrolase activity, is expressed in the apical side of the prostatic ducts and is upregulated by PCa cells. The binding of the PSMA ligand to its anchored cell membrane target mediates internalization through clathrin-dependent endocytosis, thereby enhancing the retention of conjugated radionuclides into the cells, even in small-volume sites [10,11]. The first radiometalated PSMA agent, technetium-99m, a labeled inhibitor of PSMA, was developed in 2008 [12], and later in 2010, the first ^68^Ga-labeled PSMA inhibitor was synthesized as the first clinical agent for PSMA PET imaging, which gained popularity due to its high diagnostic accuracy. It can detect as many as 50% of patients with low serum levels of prostate-specific antigen (PSA < 0.5 ng/mL) and 60% of patients with moderate levels (0.5–1 ng/mL) [9]. Moreover, it can detect additional diseases that are not detected by choline imaging [13].

Researchers have developed several chelators for ^68^Ga PSMA, including the HBED-CC chelators, known as PSMA-11 [14] and ^18^F-DCFPyl [15], which have been widely used in recent years, especially in Europe and the USA [16]. The high levels of radiotracer excretion and urinary bladder activity result in uptake in other sites other than the prostate, including the salivary glands and lacrimal glands, kidneys and liver, spleen, small intestine, and urinary collecting system, which could mask small local recurrences in this vicinity or be misdiagnosed as metastasis [11]. The targeted radionuclide chelators PSMA-617 (^177^Lu-labeled ligand) [17] and ^18^F-PSMA-1007 were developed with less urinary excretion than ^18^F-DCFPyl [18].

Following the rapid development of PSMA PET imaging (in no longer than a decade), hundreds of cohort and clinical trials with promising results have been published annually on its diagnostic accuracy in the diagnosis and staging of the disease and metastasis in patients with primary/recurrent PCa [19,20]. Further meta-analysis studies and prospective and randomized clinical trials also confirmed the high positive and negative predictive values (PPV and NPV) of PSMA PET for PCa [21,22,23,24]. Considering the controversy and the lack of evidence to make a definite conclusion about the appropriateness of MDT based on the PSMA PET/CT results, this review study was performed. This study also aimed to outline the differences in the treatment strategies and patient outcomes when diagnosed as LN-positive or -negative based on PSMA PET/CT.

## 2. Nodal Metastasis Detection by PET Scan

Considering the nodal involvement, some studies have elucidated a high diagnostic accuracy (sensitivity of 81.7%, specificity of 99.6%, PPV of 92.4%, and NPV of 98.9%) for the detection of LNs > 3 mm by PSMA PET in primary/recurrent PCa [25,26]. The comparison of the ^68^Ga-PSMA PET results with those of CT using LND confirmation showed higher diagnostic accuracy for PET (sensitivity, specificity, PPV, and NPV of 84%, 82%, 84%, and 82%, respectively) compared to CT (65%, 76%, 75%, and 67%, respectively) [27]. Nevertheless, a lower diagnostic accuracy (sensitivity of 40–68%) has been reported for the detection and staging of small LNs (<3 mm), which is below the spatial resolution of the PET cameras in primary/recurrent PCa [28,29]. However, compared to other diagnostic methods (CT scan, PSA level, and the Gleason score), PSMA PET still has had a higher diagnostic accuracy for biochemical recurrence (BCR) after radical prostatectomy (RP) [30]. 

The evaluation of nodal metastasis is specifically important in post-treatment follow-up to estimate the risk of metastasis. About one third of patients with treated PCa develop (biochemical) recurrence, defined as two consecutive elevated PSA levels (>0.2 ng/mL) 6–8 weeks after surgery or radiation therapy [21]. There have also been high NPVs for local LN metastasis [31] and high diagnostic accuracy for the detection of LN involvement after treatment (recurrence), even in patients with low PSA levels, in whom the metastases cannot be predicted by the biochemical assessment [32]. About 30% of patients that were unsuspected to have disease recurrence were detected by the PSMA PET/CT value [33], and in more than half, there was the N and M upstaging of LNs [34,35]. This change in diagnosis necessitates more rigorous treatment in patients, resulting in a change in the therapeutic plan of more than half of patients with PCa [11,22,36]. Considering the advantages of PSMA PET/CT scanning, especially in detecting and staging LN metastasis, this imaging method has been widely used to diagnose and determine the appropriate treatment for patients with primary/recurrent PCa [37]. It has also been shown to change the target volume (e.g., the additional irradiation of LN or bone metastasis) [38]. However, this imaging method has several limitations and pitfalls, including changes in visibility by using hormonal therapies (androgen deprivation therapy (ADT)). This is because of its negative effects on the visibility of PSMA PET/CT, which must be considered during the performance and interpretation of the results [39,40]. Furthermore, the long-term follow-up of the patients demonstrated that the patient outcomes were not as acceptable as previously thought, thereby questioning the appropriateness of metastasis-directed therapy (MDT) alone for patients with RPCa [41]. Another important issue undermining the use of MDT for LN metastasis is the risk of failed treatment and recurrence after the treatment of the LN metastasis [42], which increased the mortality rate (>60%) [41]. This was found while most studies considered the diagnostic accuracy of PSMA PET/CT and its impact on the change in PCa management without valid follow-up results to understand its effect on the clinical outcomes and patient prognosis [43,44]. Accordingly, the results of studies on patient outcomes after treatment decisions using PSMA PET/CT scan results are described below to shed some light on this issue.

## 3. Nodal Irradiation in PSMA-Positive Patients

Many studies have suggested a change in the therapeutic plan of a given patient, especially considering the treatment for LN involvement, according to PSMA PET/CT scan results [11,33,36]; however, most studies have reported the overall rates and have not reported the patient outcomes with PSMA-positive or -negative LNs. The treatment options for patients with localized or regional LN metastases of PCa include surgery (LND) or irradiation (LNRT, with or without androgen deprivation). There is controversy related to the preference for these methods in PSMA-positive LNs in patients with the recurrence of PCa after RP. 

A systematic review of 27 studies (6 of which used PSMA) showed a mean complete biochemical response in 44.3% (13–79.5%) of cases after SLND with 2- and 5-year biochemical-recurrence-free survival rates (BCRFS) of 23–64% and 6–31%, respectively [45]. The inconsistencies in the reported patient outcomes could be caused by differences in the type of detection method used, the type of RT (LN-specific stereotactic body RT (SBRT) or whole-pelvis radiotherapy (WPRT)), and the variation in the RT regimens and doses [46]. In a recent study involving 100 patients with BCR (24%) or biochemical persistence (76%), the results showed that PSMA PET/CT could detect 1, 2, 3, or more LN metastases in 35%, 23%, and 42% of the patients; the treatment of all LN cases with RT and ADT in 83% of patients showed improved BCR-free survival in these patients, thereby confirming the use of RT based on the results of a PET/CT scan [47]. Furthermore, another recent study showed that most patients with high PSA levels considered that a relapse of PCa could be successfully diagnosed as a recurrence by ^68^Ga-PSMA (63%). This study also concluded that PSMA PET/CT scans have a high level of significance in predicting the outcomes of patients with PSA relapse [48].

The table below summarizes the results of studies evaluating the outcomes of patients with PSMA-positive LN, organized in chronological order (Table 1). As the present review focuses on the PSMA PET/CT, those studies using a different diagnostic method (such as choline or FDG PET/CT, etc.) were not included. We also evaluated the results of the studies addressing the patient outcomes after the diagnosis of LN by PSMA PET/CT and did not include studies reporting the results of radioguided treatment strategies using this imaging modality.

Porres et al. investigated the outcomes of radiation in patients with BCR and PET-positive LNs (^18^FEC or ^68^Ga-PSMA). In a seven-year study involving 87 patients, 87.4% of the cases had undergone RP, 57.9% of the patients had adjuvant/salvage RT (additionally), and 18.4% of the participants received ADT before sLND. The patients’ favorable outcomes implied that extended salvage lymph node dissection is an appropriate and safe therapy in these patients, which allows for the postponement of systemic therapy [49]. A study involving 23 patients with PET-positive LNs also showed that RT significantly decreased PSA levels from the median of 2.75 to 1.37 ng/mL. The researchers concluded that RT is a promising therapy for the local treatment of patients with an isolated LN metastasis of PCa [50]. In an extensive multi-institutional analysis of patients with BCR and PET-positive LNs (^11^C-choline or ^68^Ga-PSMA), Fossati et al. showed that the patient outcomes after salvage LND depend on the clinical recurrence rate. According to their findings, they developed a model to predict the early clinical recurrence one year after salvage LND according to the Gleason score, the time from RP to PSA rising, hormonal therapy at PSA rise after RP, retroperitoneal or three or more spots on a PET/CT scan, and the PSA level at SLND. These researchers suggested the use of this tool for appropriate patient selection [51]. As shown above, the existing evidence indicates that PET-positive LNs are an appropriate diagnostic tool for the definite diagnosis of PCa recurrence. Other researchers have also confirmed ^68^Ga-PSMA PET/CT for the diagnosis of positive pelvic LNs in patients with BCR or high-risk primary PCa (one false-negative LN and two false-positive LNs) [54] (this study was not included in the table since the table only addresses studies on RPCa).

Few studies have compared the patient outcomes for different treatment modalities. Schmidt-Hegemann et al. compared the results of 67 patients who underwent salvage LNRT with 33 patients who underwent salvage LND and reported the priority of LNRT, considering the lower rates of distant metastasis (92% vs. 30%), the need for secondary treatments (5% vs. 39%), and prolonged BCRFS (HR = 4.204) [52]. In 2021, Kretschmer et al. compared the outcomes of 71 patients undergoing salvage LND with 67 patients undergoing salvage LNRT and reported similar MFS, general health-related quality of life, daily pad usage, and scores for the two modalities. However, the only significant difference was associated with a higher PSA-progression-free survival in the salvage LNRT group [53]. 

An important issue challenging the comparison of the rates of patient outcomes among studies is the presence of confounders, i.e., factors affecting the patient outcomes that are independent of the treatment plan or the diagnostic accuracy of the PSMA PET/CT scan. These factors include the number of positive LNs on the ^68^Ga-PSMA PET/CT scan, the Gleason score, the duration of ADT before recurrence, and the duration from the initial diagnosis to relapse [48,55]. Fossati et al. have also developed a model to predict the outcome of salvage LND according to Gleason grade group 5, the time from RP to PSA rising, hormonal therapy at PSA rise after RP, retroperitoneal or three or more spots on a PET/CT scan, and the PSA level at SLND [51]. 

Interestingly, Farolfi et al. compared the results of ^68^Ga-PSMA PET/CT before and after salvage LND in 16 patients with persistent BCR and determined the recurrence after LND in 25% of cases (*n* = 4) and repeated local therapy after salvage LND in 9 patients (7 with RT and 2 with surgery). They also reported that all regions detected by PET as positive were truly positive [56]. These findings suggest that the selected MDT was not a complete treatment. Considering the high mortality rate in patients with failed salvage therapy (above 60%) [41], it is important to select an appropriate treatment method to reduce the risk of failed treatment and recurrence after the treatment of LN metastasis [42]. De Bari et al. suggested that adopting larger target volumes treated at least 95% of lymph node regions with the risk of occult relapse [57]. It was also suggested to estimate the oncologic benefit of MDT and select the most appropriate treatment strategy regarding patients’ conditions when this imaging tool was used for treatment decisions in LN-positive patients [51]. 

By the accumulation of the above data, in one of the most important ongoing trials the investigators are testing the benefit of treating PET- and/or MRI-defined involved nodes by IMRT or SBRT along with the elective treatment of the pelvic nodes and the prostatic bed in the salvage settings [58]. 

## 4. Nodal Irradiation in PSMA-Negative Patients

In the recently published SPPORT study, the elective treatment of the pelvic nodes in the pre-PSMA era has been associated with superior BCRFS compared to treatment of prostate bed alone in the salvage setting [59]. However, the treatment and/or outcome of patients diagnosed as LN-negative using a PSMA PET/CT scan has mainly been reported in the subgroup analyses of studies and scarcely as their main objective. Most studies have indicated that patients with PSMA-negative LNs had a lower PSA compared to those with PSMA-positive LNs [32], suggesting that they have a better prognosis (1- and 2-year BCRFS rates of 87% and 76%, respectively) [60]. Comparing the results of histological metastasis, as determined by LND, with the results of PSMA PET also showed that the consideration of a negative 68Ga-PSMA PET/CT as the basis of not performing pelvic LND can avoid unnecessary LND treatments in 80% of patients [61]. However, the specificity of PSMA PET/CT in predicting pathologically confirmed positive nodes ranged from 87.5 to 97.3%, and only 24% of patients diagnosed as negative by PSMA were found to be positive histologically [45]. Comparing the histological reports with the PSMA PET results in patients before salvage LND showed specificity values of 74.1% in the side-based analysis and 87.5% in the LN field-based analysis and an NPV of 90.9% in the LN field-based analysis [62]. Nonetheless, the majority of studies have not reported the NPV of PSMA PET/CT in patients with RPCa [22,63], in which has been reported in patients with primary PCa [64,65].

As discussed earlier, small LNs cannot be captured by PET scans, and comparing the results of 68Ga-PSMA PET/CT or PET/MRI with histopathologic results showed a median diameter of 3.4 mm (IQR 2.1–5.4 mm) for metastatic LNs that were considered negative on a ^68^Ga-PSMA PET/CT scan [66]. Furthermore, although the recurrence rate of PSMA-negative patients is lower than those with positive PSMA (16.7% vs. 50%, respectively) [30], some have reported similar BCRFS rates between PSMA-negative and -positive patients (82% vs. 74%, respectively) [67]. Accordingly, the BCR rate in PSMA-negative patients should be considered. A follow-up of 103 patients with BCR and negative PSMA LNs who were receiving no treatment detected clinical recurrence in the prostatic fossa (45.6%), nodes (38.6%), and bone (15.8%) at a median of 15.4 months, with overall clinical-recurrence-free survival rates of 61.4% after one year and 34.8% after two years, which was longer in patients with a lower ISUP grade group [68]. These findings support the necessity of active surveillance for these patients using on-time and appropriate therapeutic strategies. However, leaving these patients without treatment may be a great risk; some suggest salvage LNRT, even in the absence of PSMA-diagnosed LNs, considering the low sensitivity of PSMA PET/CT in diagnosing micrometastases [38,69]. The Table 2 summarizes the results of studies reporting the outcomes of PSMA-negative LNs in patients suspected of BCR.

Emmet et al. focused on the predictive value of negative PSMA results for LN metastasis in patients with persistent PSA (with PSA readings between 0.05 and 1.0 ng/mL). In their first study (2017) among 60 patients with a negative PSMA result, 27 patients underwent SRT (45%), and the others did not; among those not receiving treatment, 65% had increased PSA levels. They also showed a high response to treatment in PSMA-negative patients, highlighting the value of treatment in these patients [71]. In 2021, they published the results of a three-year follow-up of 260 patients. In their study, 32% of PSMA-negative patients did not receive treatment, and 66% showed PSA progression (with a mean rise of 1.59 ng/mL over three years). They also reported the higher likelihood of salvage LNRT and ADT in PSMA-positive patients compared to PSMA-negative patients [72]. Zschaek et al. evaluated patients with extremely high risk PCa who underwent PSMA PET before salvage LNRT and showed that treatment with salvage LNRT significantly prevented PSA increase in patients with negative PSMA LNs [70]. Schmidt-Hegemann et al. evaluated 204 consecutive patients that were referred for salvage LNRT and underwent PSMA PET before treatment; about half of their study population had negative PET results, 81% of whom also had a low PSA level (≤0.5 ng/mL). None of the patients with a negative PET result underwent LNRT or other treatments (only one continued ADT). However, the results showed that the PSMA results (positive or negative) did not influence the outcome (BFRS), which was mainly due to the advantage of treatment intensification in patients with positive PSMA PET results [67]. A review of 27 studies (*n* = 2832 patients with a primary diagnosis of PCa) also confirmed that the patient risk score should be considered for the decision of pelvic LND, even in patients with negative PSMA PET/CT results [73]. Other studies have also concluded that a negative PSMA PET/CT result does not rule out LN metastasis [29,74]. Accordingly, it is speculated that a risk scoring system should be used for making decisions about the treatment of PSMA-negative patients in cases with BCR of PCa. However, such results have not been confirmed for these patients. Kiste et al. reported their results in their cohort based on PSMA-negative PET results and showed that the initial T status, the M status at recurrence, the PSA level at the time of salvage LNRT, additive ADR, and elective prostate bed RT could significantly predict the BCRFS during a median follow-up of 28 months [75]. Further studies are required to determine the most appropriate type of RT, the extent of irradiation in patients with BCR of PCa, and no evidence of LN metastasis on a PSMA PET/CT scan. In the following section, we review the available evidence on the extent of LNRT in RPCa patients. 

## 5. Extent of the Nodal Irradiation Target Volume in RPCa Patients

We discussed the role of PSMA PET/CT scans in the definite diagnosis of recurrence. The next step after diagnosis is treatment. RT is one of the fundamental treatments in RPCa patients, for which determining the extent of RT of the LNs, which may include the external iliac, internal iliac, obturator, and presacral nodal groups, is of the utmost importance. However, the extent of RT is controversial. On the one hand, failure to identify the microscopic disease reduces the effectiveness of RT and increases the recurrence rate.

On the other hand, extensive RT to the prostatic fossa can result in urinary and bowel morbidity due to the unnecessary irradiation of normal tissues. Accordingly, it is necessary to accurately target areas harboring subclinical tumors and identify the extent of irradiation for intrapelvic target volumes (the prostate, seminal vesicles, prostatectomy bed, and lymph nodes) and other organs at risk (e.g., the bladder and rectum). This requires planning the clinical target volume (CTV) margins. There is much controversy about this issue in patients with primary PCa [76], and little evidence is available in this regard in RPCa patients.

There are four distinct consensus guidelines that have been developed by major institutions (multidisciplinary groups of health care professionals) for CTV and planning target volume (PTV) margins. Some examples are Princess Margaret Hospital (PMH), the Australian and New Zealand Faculty of Radiation Oncology Genito-Urinary Group (FROGG), the European Organization for Research and Treatment of Cancer (EORTC), and the Radiation Therapy Oncology Group (RTOG). The initial contours were generated by the PMH Consensus Workshop on Postprostatectomy Radiotherapy in June 2006 and were modified by the FROGG in their consensus atlas [77]. A detailed description of the methods involved in the creation of the contouring guidelines was published elsewhere [78]. The 2010 consensus of the RTOG on the prostate fossa clinical target volume (PF-CTV) of RT after RP defined using the “vascular expansion” technique, which is currently used by several specialists. It suggested an inferior border at least 8–12 mm below the vesicourethral anastomosis (VUA) and indicated that the superior limit would not extend more than 3–4 cm above the pubic symphysis unless there was a gross disease or a seminal vesicle remnant. The urinary bladder and bladder neck (with the circumference of the wall) are suggested to be included in the superior aspect of the retropubic component of the CTV [76]. 

Treatment failure using the above guidelines suggested the need for its modification or alteration, for which the focus of attention has been placed on the imaging method. A comparison of CT and MRI images showed that a contouring CTV based only on MRI depicts the prostate apex and its intersection with the bladder better and could spare more of the bladder wall than the RTOG CTV [79,80]. The development of better imaging tools, including molecular imaging, has revealed promising results in metastatic, locally advanced, localized, and postprostatectomy PCa, which can also be used to define the CTV in RPCa patients. In the study by Jani et al., the modification of the postprostatectomy CTV by adding ^18^F-fluciclovine PET/CT resulted in a larger CTV (especially on the prostate bed and pelvis volumes), while it did not result in remarkable toxicity on the rectum and bladder, suggesting the appropriateness of using this imaging method for a better CTV definition [81]. However, the authors concluded that more longitudinal studies are required to reach definite results. Moreover, choline PET/CT revealed 36.8% of PET-positive LNs outside the standard CTV [82]. Considering the higher detection rate of ^68^Ga-PSMA PET/CT in the diagnosis of RPCa [83], its accuracy in RT contouring is supposed to be higher. However, scant evidence is available to support this hypothesis.

The results of evaluating recurrence in patients with BCR using ^68^Ga-PSMA PET/CT showed that there were 32/40 with a positive LN, 25% inside, 68.75% outside, and 6.25% both inside and outside the CTV according to RTOG guidelines. These findings suggested that larger target volumes were required for targeting the occult relapse [57]. In another study on 15 patients, using ^68^Ga-PSMA PET/CT changed the TNM staging in 53% and the RT concept, including the clinical and gross target volumes (GTV), in one third of the patients [34]. Schiller et al. designed a three-dimensional atlas for PSMA-PET/CT-based LN metastases and showed that the standard RTOG CTV was only accomplished in 31% of all LN metastases; the major uncovered areas included the para-aortal, pararectal, paravesical, preacetabular, presacral, and inguinal regions [84]. Furthermore, another study using ^68^Ga-PSMA PET/CT showed that 30% of LNs were outside the standard RT CTV intensity-modulated radiation therapy (IMRT), especially in cases with higher levels of PSA [85]. Accordingly, one should note that the pattern of the LN involvement of RPCa patients differs from those with primary cancer, and the anatomical boundaries of RT outlined based on the surgical clips [76,86] are not applicable here. Therefore, newer guidelines have been developed for the target delineation of LNs for salvage RT as a reflection of the changes made by PSMA PET/CT in RPCa patients [87]. 

Among the studies evaluating RPCa using PSMA PET/CT scans reviewed in Section 2 and Section 3, in Table 3, we summarized those using RT as a salvage treatment and described the the details of the RT target volume, dose and toxicity, in chronological order.

Henkenberenz et al. clearly defined GTV, CTV, and PTV in their study (considered based on pathological findings in the planning CT, low-dose CT of the ^68^Ga-PSMA PET/CT, and MRI) and showed that 90% of LNs that were found to be positive on PSMA PET/CT were outside the radiotherapy fields, and the applied RT resulted in the local control of at least 95.6% after a median follow-up of 12.4 months (no pathological tracer uptake) [50]. Although these findings confirm the appropriateness of the ^68^Ga-PSMA PET/CT in determining the RT extent in RPCa patients, we cannot compare the rates reported for the LNs outside the standard CTV between the studies, as each considered a different consensus guideline as the standard and some were not mentioned in their published article, such as the previously mentioned study [50]. Like Kretschmer et al. [53], Schmidt-Hegemann et al. considered the RTOG standards in both of their studies [52,67]; however, they did not report the number of LNs outside the CTV field. A notable finding in these studies was the high rates of toxicity grades II and III, acute and late. Nonetheless, compared with SLND, more favorable results were reported for SLNRT in RPCa patients [52,53]. Comparing the outcomes of SBRT, which treats only the PET-positive LNs, with WPRT, which treats the whole lymphatic drainage (the entire pelvic lymphatic pathway) and the prostate itself, has shown a higher relapse rate after SBRT, particularly in the pelvic lymph nodes. However, ENRT was associated with higher late toxicity and a greater need for hormonal therapy [88]. In contrast, most studies have not clearly defined the contouring of RT or the involved LNs, and the subgroup of patients with positive LNs was too small to draw a conclusion (47, 70–72). Accordingly, more evidence is required on the appropriateness of using ^68^Ga-PSMA PET/CT to determine the RT extent in RPCa patients.

## 6. Conclusions

The present review study focused on the role of PSMA PET/CT in diagnosing RPCa. Routinely, patients with increased PSA levels after RP are considered at high risk for recurrence. The first step in assessing these patients would be ruling out distant metastases that dichotomize. The novel development of PSMA PET/CT has facilitated the more accurate determination of LN involvement in these patients. Although several studies have outlined the diagnostic accuracy of this imaging modality, little is known about its role in patient outcomes. In this review study, we categorized treatments based on the positivity/negativity of PSMA PET/CT, indicating that salvage LNRT is an appropriate method for treating cases with positive LNs on PSMA PET/CT; however, the treatment of PSMA-negative LNs is still controversial.

Furthermore, the few studies prescribing LNRT for these patients did not clearly define the extent of the RT volume, which significantly affects patient outcomes. The existing guidelines have mainly focused on RT nodal target volumes in patients with primary PCa and not in the salvage setting. Hence, most the appropriate extent of nodal target volumes in these patients needs to be defined. Given the treatment of patients with relapse after the treatment of RPCa, adopting larger target volumes to treat more LNs may seem wise. However, further studies should be conducted to define the extent of RT in RPCa patients according to PSMA PET/CT imaging.

## 7. Future Perspective

With the evolution of prostate-specific PET/CT scans, newer tracers other than the ^68^Gallium-PSMA PET/CT scan are being introduced with higher sensitivity and specificity in finding the disease at lower PSA levels and are enabling curative intent for recurrent tumors. One of these tracers is 18F- PSMA, which has shown promising false-negative rates and high-quality imaging properties of itself. In addition, higher-resolution detectors are being developed that will show the uptake of radionuclides in smaller LNs that are currently missed by the available scans. Another proposition by imaging specialists is using a PET-MRI modality that uses MRI instead of a CT scan, which gives a superior resolution in the soft tissue and fat composition of the pelvis. 

Aside from the imaging improvement, it is hoped that more successful treatment, lower morbidity, and longer OS will be achieved by these newer targeted radioisotopes detecting a higher percentage of diseases at lower PSA levels (<0.2) or in micrometastatic LNs (<3–4 mm). The necessity of patient selection for the treatment of PSMA-negative LNs is an active area of investigation that will be clearer in the future. Currently, some stratification models have been developed to predict the risk of failure in these patients. These models need to be merged with recently found molecular prognostic factors in the context of studies with long-term follow-up to be used as a standard risk estimation model in routine practice. Another critical issue is the recurrence pattern in the PSMA-PET-negative patients, illustrating some recurrences outside the routine radiation target volume. Accordingly, larger radiation volumes may be required to reduce treatment failure. Further studies may suggest more modifications based on PET/CT as the standard of care in the target delineation of PCa. 

## 8. Executive Summary or Practice Points


The conventional imaging methods, including MRI and CT scanning, have limited diagnostic accuracy for LN involvement in patients with PCa. In the past few years, the development of PSMA PET, with high positive and negative predictive values, has transformed the diagnostic and therapeutic approaches to PCa. Although, in the case of negative PSMA PET results and positive MRI results, we should take the more conservative action and identify the given node as malignant.Several studies have suggested changes in the therapeutic plans of patients, especially in cases with LN involvement, according to the PSMA PET/CT scan results. The involved nodes may change the target volume of RT or the dissection field.The treatment of all LNs detected on PSMA PET using RT and ADT showed improved BCR-free survival in 83% of patients, thereby confirming the selection of treatment based on the PET/CT scan results. When irradiating these cases, it would be best to dose-escalate the positive node as much as safely possible, considering the availability of IGRT using IMRT/VMAT or SBRT techniques. This approach is currently being applied in the PEACE-V-STORM trial.Patients with PSMA-negative LNs had a lower PSA than those with PSMA-positive LNs, suggesting a better prognosis in this group. However, the likelihood of salvage LNRT and ADT was higher in the PSMA-positive patients compared to the PSMA-negative patients.Negative ^68^Ga-PSMA PET/CT, as the basis for not performing pelvic LND, may avoid unnecessary LND treatments in about 2/3 of the patients since only 24% of the PSMA-negative patients were found to be positive histologically.Leaving patients with BCR and PSMA-negative LN without treatment may harbor a remarkable risk of disease progression since in a some of the patients clinical RPCa was detected in the prostatic fossa (45.6%), nodes (38.6%), and bone (15.8%) during a median follow-up of 15.4 months. It is encouraged to use a risk scoring system for decision-making about the treatment of PSMA-negative patients. This system routinely includes the primary T status and the extent of LND and positive nodes at the primary surgery, the PSA doubling time, and GS.Strong responses to treatment (salvage LNRT) and increased PSA levels in 65% of PSMA-negative patients highlight the value of treatment in these patients. This evidence suggests the low sensitivity of PSMA PET/CT in the diagnosis of micro-metastasis (<3–4 mm LNs). The availability of newer and safer RT techniques and the results of the recently published SPPORT trial encourage more radiation oncologists to electively irradiate the clinically negative nodes based on the new proposed LN delineation guidelines. This treatment has a high efficacy and a very low toxicity.The results of evaluating RPCa in patients using ^68^Ga-PSMA PET/CT showed a positive LN outside the CTV in 30–68.75% of cases, according to the RTOG guidelines. These findings suggest larger target volumes are required for targeting the occult relapse. The major uncovered areas included the para-aortal, perirectal, paravesical, preacetabular, presacral, and inguinal regions.Note that the pattern of the LN involvement of patients with RPCa differs from those with primary cancer; hence, guidelines should be provided to define the RT extent in RPCa patients.Newer radionuclide tracers are currently under development that could show the microscopic disease in small LNs and with lower PSA levels. These tracers have yet to become standard in routine clinical practice.


## Figures and Tables

**Table 1 biomedicines-11-00038-t001:** The summary of studies reporting the treatment outcomes of PSMA-positive lymph nodes in patients with a recurrence of prostate cancer after primary treatment with radical prostatectomy.

Author, Year of Publication	Number of Patients	Imaging Method	Treatments Applied	Median Follow-Up (Months)	Conclusion	
					Recurrence or Response Rate	Survival
Porres et al. [49], 2017	87	^18^FEC or ^68^Ga-PSMA PET/CT	Salvage extended lymph node dissection	21	Complete biochemical response: 27.5%, Incomplete biochemical response: 40.6%	ADT-free: 62.2%, CSM: 3.7%, 3-year BCR-free: 69.3%, systemic-therapy-free survival: 77.0%, clinical-recurrence-free survival: 75%, for patients with complete biochemical response
Henkenberenz et al. [50], 2017	23	^68^GA-PSMA PET/CT	Salvage LNRT	12.4	Recurrence outside the initial radiation field: 12.9%	BCR-free survival: 95.6%, systemic-therapy-free survival: 100%
Fossati et al. [51], 2019	654	^11^C- or ^68^Ga-PSMA PET/CT	Salvage LND	30	Early clinical recurrence: 25%	CSM: 20% in patients with and 1.4% in patients without early clinical recurrence
Schmidt-Hegemann et al. [52], 2020	100	^68^GA-PSMA PET/CT	Salvage LNRT vs. LND	17 in SLND and 31 in salvage LNRT	LND had higher distant metastases (52% vs. 21%) and secondary treatments (39% vs. 15%).	2-year BCR-free survival was 92% in salvage LNRT and 30% in SLND
Kretschmer et al. [53], 2021	138	^68^GA-PSMA PET/CT	Salvage LNRT vs. LND	47 in SLNRT and 31 in SLND	BCR: 40.3% for SLNRT and 86.4% for SLND, distant metastasis: 31.3% for SLNRT and 36.4% for SLND	Median metastasis-free survival: 70 months for all (57.6 months for SLNRT and 39.5 months for SLND; not different)
Rogowski et al. [47], 2021	100	^18^FEC and ^68^Ga-PSMA PET/CT	sENRT	37	Metastasis: 83% only pelvic, 2% only para-aortic, 15% pelvic and para-aortic LN metastases.	1, 2-, and 3-year BCR-free survival: 80.7%, 71.6%, and 65.8%, and 1, 2-, and 3-year distant-metastasis-free survival: 91.6%, 79.1%, and 66.4%, respectively

Abbreviations: BCR, biochemical recurrence; ADT, androgen deprivation therapy; CSM, cancer-specific mortality; RP, radical prostatectomy; LNRT, lymph node radiotherapy; PSMA PET/CT, prostate-specific membrane antigen positron emission tomography/computer tomography; LND, lymph node dissection; sENRT, salvage elective nodal radiotherapy; PSA, prostate-specific antigen.

**Table 2 biomedicines-11-00038-t002:** The summary of studies reporting the treatment outcomes of patients with PSMA-negative results for prostate bed and lymph node metastasis in patients with the biochemical recurrence of prostate cancer (referred for treatment) after radical prostatectomy.

Author, Year of Publication	Ratio of Patients with Negative PSMA PET/CT to All Patients	Treatments Applied	Median Duration of Follow-up (Months)	Conclusion
Zschaek et al. [70], 2017	Not mentioned	Salvage LNRT	29	Median PSA response:9% decline for PSMA-negative patients and pathological N+ vs. 79% decline for PSMA negative and pathological N0
Emmett et al. [71], 2017	60/164	Salvage LNRT and prostate bed RT	10.5	In total, 85% with negative PSMA responded to treatment, and PSA increased in 65% of untreated patients.
Schmidt-Hegemann et al. [67], 2019	48/90	Salvage LNRT and/or ADT	23	Similar recurrence-free rates between positive and negative PSMA (74% vs. 82%)
Emmett et al. [72], 2020	90/260	Salvage LNRT and prostate bed RT		Negative PSMA plus salvage LNRT was the best predictor of 3-year free-from-progression rate (82.5%), and 66% of untreated patients had a PSA increase.

Abbreviations: PSMA, prostate-specific membrane antigen; LNRT, lymph node radiotherapy; PSA, prostate-specific antigen; RP, radical prostatectomy. Note: all studies used ^68^GA-prostate-specific membrane antigen positron emission tomography/computer tomography as the imaging method.

**Table 3 biomedicines-11-00038-t003:** The details of radiotherapy target volume, dose and toxicity in studies that used radiotherapy for the treatment of prostate cancer recurrence based on PSMA PET/CT findings.

Author, Year of Publication	No. of Patients Who Received RT	Treatments Applied	Definition of Target Volumes	Lymph Nodes Involved	Radiotherapy Details	Toxicity
Zschaek et al. [70], 2017	20	Salvage LNRT (IMRT) only to those with positive PET/CT	in case of negative PSMA PET: Prostate bed and seminal vesicles. If positive PSMA added pelvic nodes	Not mentioned	Prostate bed irradiated to 66.6 Gy. If positive margins or extra-capsular invasion a SIB to 70.3. if positive PSMA in prostate bed increasd dose to 74–77 Gy. LN drainage sites received 54.0 Gy, while macroscopic LNs on PET received 66 Gy.	Well tolerated, 2 cases of >grade I acute toxicity (grade II noninfective cystitis and diarrhea)
Henkenberenz et al. [50], 2017	11	Salvage LNRT	GTV based on CT, PSMA PET, and MRI. CTV as the area with pathological tracer uptake; PTV with 10 mm safety margins in all directions around CTV	Para-aortic and retroperitoneal (54.5%), mediastinal (18.2%)	LNRT included five times weekly with 2.0 Gy up to a total dose of 50.4–54.0 Gy	No grade III acute or grade II late toxicity, 21.7% had grade II diarrhea and 8.7% had persistent grade I diarrhea, no deterioration of urinary or fecal continence
Emmett et al. [71], 2017	99	Salvage LNRT and the prostate bed RT	Three categories of RT: Prostate fossa-only, Prostate fossa + pelvic nodes, or SBRT external to the pelvis.	fossa + pelvic nodes	Not mentioned	Not mentioned
Schmidt-Hegemann et al. [67], 2019	18	Salvage LNRT (IMRT or image-guided VMAT) ± ADT	PTV was considered the 5–7 mm expanded CTV margin in all directions	13% pelvic LNs, 7% fossa + pelvic LNs	Total of 45–50.4 Gy, with simultaneous or sequential boost	Grade II genitourinary and gastrointestinal toxicity were present as 13% and 16% acute and 13% and 3% late, 2 patients with grade III late genitourinary toxicity
Schmidt-Hegemann et al. [52], 2020	67	Same as above	Same as above	Same as above	Same as above	Acute grade II gastrointestinal and urogenital toxicity in 28% of patients, acute grade III urogenital toxicity in 2%, late grade II toxicity in 36%, and grade III in 37%
Emmett et al. [72], 2020	186	Salvage LNRT (25% with ADT)	Not mentioned	49.4% to the fossa + pelvic LNs, 12.4% LNs or stereotactic body	Not mentioned	Not mentioned
Kretschmer et al. [53], 2021	67	Salvage LNRT (IMRT or VMAT) +ADT (*n* = 61) vs. LND	RTOG	Not mentioned	Median of 61.6 Gy (range: 50.4–66 Gy).	Acute grade II gastrointestinal and urogenital toxicity in 28.4%, acute grade III urogenital toxicity in 1.5%, late grade II toxicity in 35.8%, and grade III in 37.3%
Rogowski et al. [47], 2021	100	Salvage LNRT (IMRT or VMAT) +ADT	RTOG	83% only pelvic, 2% only para-aortic, and 15% pelvic + para-aortic LNs	Median 65.1 Gy (Range: 56–66 Gy)	Not mentioned

Abbreviations: PSMA PET/CT, prostate-specific membrane antigen positron emission tomography/computer tomography; LNRT, lymph node radiotherapy; SBRT, stereotactic body radiation therapy ; IMRT, intensity-modulated radiotherapy; VMAT, volumetric modulated arc therapy. Toxicity was defined based on the Common Terminology Criteria for Adverse Events (CTCAE) Version 4.0.

## Data Availability

Not applicable.

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
