# Peer review of "Necessity of Pelvic Lymph Node Irradiation in Patients with Recurrent Prostate Cancer after Radical Prostatectomy in the PSMA PET/CT Era: A Narrative Review"

_biomedicines, 2022, doi:10.3390/biomedicines11010038_

Round 1
Reviewer 1 Report
The subject of this Review article "Necessity of Pelvic Lymph Node Irradiation in Patients with Recurrent Prostate Cancer after Radical Prostatectomy in the PSMA PET CT Era; a Narrative Review" has a considerable clinical impact.
However, this paper is poorly written in some parts and the entire manuscript needs an intensive English correction.
Furthermore, there are incorrect information concerning some topics such as the definition of biochemical recurrence of prostate cancer treated with radical prostatectomy or radiotherapy. In addition, the article lacks a robust conclusion.
Author Response
The subject of this Review article "Necessity of Pelvic Lymph Node Irradiation in Patients with Recurrent Prostate Cancer after Radical Prostatectomy in the PSMA PET CT Era; a Narrative Review" has a considerable clinical impact.
However, this paper is poorly written in some parts and the entire manuscript needs an intensive English correction.
- Thanks, a comprehensive English review was done.
Furthermore, there are incorrect information concerning some topics such as the definition of biochemical recurrence of prostate cancer treated with radical prostatectomy or radiotherapy. In addition, the article lacks a robust conclusion.
- We did not provide any new definition for biochemical recurrence and we used those definitions used by other investigators in each study.
Reviewer 2 Report
The authors need to be congratulated on this research topic of high interest to treating physicians and conduction of their systematic review.
As a minor comment, the future perspective section needs to be expanded, however the inclusion of the Executive summary or practice point: section is brilliant.
Second, there is no clear section on methodology for this systematic review.
Third some minor English polishing is necessary throughout the manuscript.
Author Response
The authors need to be congratulated on this research topic of high interest to treating physicians and conduction of their systematic review.
As a minor comment, the future perspective section needs to be expanded, however the inclusion of the Executive summary or practice point: section is brilliant.
- Thanks, amended.
Second, there is no clear section on methodology for this systematic review.
- We followed the journal theme for writing Narrative Reviews so we did not provide a specific methodology. This was not meant to be a systematic review.
Third some minor English polishing is necessary throughout the manuscript.
- Thanks, amended.
Reviewer 3 Report
As a clinical pharmacologist and biostatistician, I have no concerns about this manuscript. Anyway, I do only reccomend to use BCR and not BR for biochemical recurrence, since it's a widely accepted definition
Author Response
As a clinical pharmacologist and biostatistician, I have no concerns about this manuscript. Anyway, I do only recommend to use BCR and not BR for biochemical recurrence, since it's a widely accepted definition
- Thanks, amended.
Round 2
Reviewer 1 Report
Well done. I have no further comments or recommendations.